# Causal Inference from Small High-dimensional Datasets

## Abstract

Many methods have been proposed to estimate treatment effects with observational data. Often, the choice of the method considers the application's characteristics, such as type of treatment and outcome, confounding effect, and the complexity of the data. These methods implicitly assume that the sample size is large enough to train such models, especially the neural network-based estimators. What if this is not the case? In this work, we propose Causal-Batle, a methodology to estimate treatment effects in small high-dimensional datasets in the presence of another high-dimensional dataset in the same feature space. We adopt an approach that brings transfer learning techniques into causal inference. Our experiments show that such an approach helps to bring stability to neural network-based methods and improve the treatment effect estimates in small high-dimensional datasets. The code for our method and all our experiments is available at `github.com/HiddenForAnonymization`.

## 1 Introduction

Recent progress in Causal Inference has led to the widespread adoption of the field. Many of these new applications require more sophisticated methods or the adaptation of existing approaches. This work proposes a treatment effect estimator which targets small high-dimensional datasets. To illustrate our approach, consider a Computational Biology (CB) application: CB applications often have detailed information from each patient, such as SNPs, genes, mutations, and parental information, but only a few patients. Hence, small high-dimensional datasets. The main bottleneck to collecting more samples is the high costs of finding new patients, plus the event's rarity. In pharmacogenomics, for instance, some applications aim to explore the side effects of drugs adopted on certain pediatric cancer treatments. Collecting information about these patients is quite challenging for several reasons: first, it requires the families' commitment in a very stressful and delicate time of their lives; second, it is a rare disease; finally, it requires a common effort between hospitals and doctors to refer the patients. While there are methods to estimate binary treatment effects (Glynn & Quinn, 2010; Hill, 2011; Shalit et al., 2017; Shi et al., 2019), and methods to estimate treatment effects with high-dimensional datasets (Jesson et al., 2020), there is a lack of methods to estimate treatment effects in small high-dimensional datasets, such as the case of several computational biology applications.

In the Machine Learning (ML) community, transfer learning is the go-to method when handling small datasets. In that case, the overall goal is to transfer knowledge from a large source-domain dataset to a small target-domain dataset. However, transfer learning for causal inference has been underexplored. In the causal inference literature, Pearl & Bareinboim (2011), Bareinboim & Pearl (2014), and Guo et al. (2021) explore the transfer of knowledge from Randomized Control Trials (RCTs) to observational data. Yang & Ding (2019) and Chau et al. (2021) focused on data fusion, which combines several datasets with a two-step estimation method or combines DAGs with a common node. At the same time, most of the existing transfer learning approaches from the ML community focus on predictive and classification tasks. When causality is used, it is usually to improve a transfer learning or domain adaptation method (Causality for ML *versus* ML for causality).

The practical usefulness of Causal-Batle is most often observed in biomedical applications. Large sets of labeled data with treatments and outcomes in biomedical applications can be expensive or impossible to obtain. Nevertheless, unlabeled source-domain data is often available. For instance, consider a study investigating an intervention on eye pressure (outcome of interest). For each patient, we have an image of

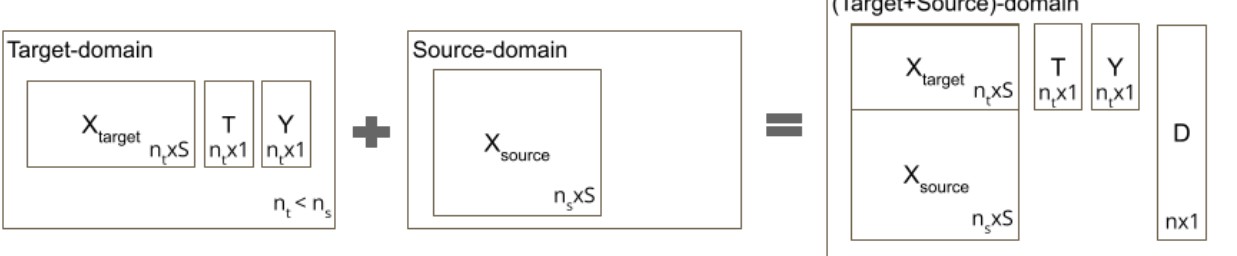

Figure 1: Illustration of how Causal-Batle combines the target and source-domain. As per the definition of small high-dimensional datasets, the number of samples in the target-domain is small($n_t$ and $n_t < n_s$) and high-dimensional (large $S$). Only the target-domain contains treatment assignment ($T$) and outcome ($Y$). In the combined dataset, the treatment $T$ and the outcome $Y$ for the samples in the source-domain are left empty. These empty values do not affect our model, as they are not used by the predictors or loss functions. The variable $D$ is binary and encodes the samples' domain, $n = n_t + n_s$ and $X = [X_{target}, X_{source}]$ (See more in Section 3).

the rear of the eye, also known as a fundus image, used to perform covariate adjustment. In this case, the dataset of such a study could be augmented by adding other fundus images as part of the source domain (images without treatment assignment or eye pressure associated). Causal-Batle introduces an approach to combine source and target domain to perform covariate adjustment and estimate causal effects.

This paper focuses on applications with a small high-dimensional dataset (target-domain), where a large unlabeled dataset is also available (source-domain). The unlabeled dataset is employed to learn a better latent representation that allows us to estimate potential outcomes in the target-domain. We assume that the target-domain and source-domain share the same feature space.

Our main contributions are as follows:

- We introduce and formalize the problem of transfer learning for causal inference from small high-dimensional datasets and discuss the underlying assumptions and the identifiability of the treatment effect.

- We propose Causal-Batle, a causal inference method that uses transfer learning to estimate treatment effects on small high-dimensional datasets.

- We validate Causal-Batle in three datasets and compare with state-of-the-art methods to evaluate if the transfer learning approach contributes to improving the estimates.

## 2 Related Work

Our work combines treatment effect estimation and transfer learning:

**Treatment Effect Estimation**: There are many methods to estimate treatment effects, each one with its own set of assumptions, targeting different types of applications. This work estimates the average treatment effect in a binary treatment setting. Starting with the most traditional methods, AIPW (Glynn & Quinn, 2010) adopts a two-step estimation approach, and BART (Hill, 2011) adopts a bayesian random forest approach. Recently, many methods have adopted Machine Learning components, like neural networks, to estimate treatment effects. TARNet (Shalit et al., 2017), CEVAE (Louizos et al., 2017), Dragonnet (Shi et al., 2019), and X-learner (Künzel et al., 2019) are some of examples of ML-based approaches. CEVAE (Louizos et al., 2017) has also adopted a variational autoencoder to estimate latent variables used to replace unobserved confounders. CEVAE and other methods (Wang & Blei, 2019) have claimed robustness to missing confounders. These methods, however, have been under large scrutiny by the community, with

several counterexamples showing their limitations and situations where they fail (Rissanen & Marttinen, 2021; Zheng et al., 2021). Considering high-dimensional datasets, Jesson et al. (2020) has suggested that bayesian neural networks tend to perform better than their regular neural network counterparts. However, the challenge is that bayesian deep neural networks require larger datasets for training, which are often unavailable.

**Transfer Learning (TL)**: TL aims to improve applications with limited data by leveraging knowledge from other related domains (Zhuang et al., 2020). Transfer knowledge between causal experiments has been previously explored as transportability of causal knowledge (Pearl & Bareinboim, 2011; Bareinboim & Pearl, 2014; Guo et al., 2021; Dahabreh et al., 2020), where the core idea is to transfer knowledge from one experiment to another. These works assume at least two experiments available with covariates, treatment, and an outcome. In practice, the transportability of causal knowledge performs an adjustment for new populations considering the causal graph and results from previous experiments. Causal data fusion (Yang & Ding, 2019; Chau et al., 2021) combines different datasets to perform the estimation of treatment effects. The datasets' fusion is incorporated into causal inference methods, either as a smaller validation set with additional information or a two-step estimation method that combines causal graphs through a shared node. Some works use semi-supervised approaches to estimate treatment effects. Chakrabortty et al. (2022) adopts a semi-supervised setup and uses unlabeled data to improve treatment effect estimation of a labeled dataset through kernels and regression models-based estimators. Harada & Kashima (2020) also proposes a semi-supervised approach based on a matching method to estimate treatment effects. Causal-Batle is similar to the semi-supervised approaches in the sense that it also adopts an unlabeled source-domain to improve the treatment effect estimations on a labeled target-domain, where the labels are the treatment and the observed outcomes. Causal-Batle adopts transfer learning techniques explored in the ML community. In the ML community, such a technique is classified as heterogeneous transfer learning (Zhuang et al., 2020). Causal-Batle is also suitable for high-dimensional and complex datasets, something that many matching and regression-based methods have struggled with in the past.

## 3 Causal-batle

The main goal of this work is to estimate treatment effects from small high-dimensional datasets in applications where a second high-dimensional dataset is available in the same feature space but without treatment assignment or outcomes. The challenge in such applications is to extract a meaningful representation from the high-dimensional covariates, given the small number of samples. Therefore, for applications with a second dataset with a shared feature space, we propose a method that performs **Ca**usal inference with **U**nlabeled and **S**mall **L**abeled data using **Ba**yesian neural networks and **T**ransfer **Le**arning (Causal-Batle).

### 3.1 Notation

Causal-Batle is a treatment effect estimator for observational studies. We assume two domains: the target-domain and the source-domain. The target-domain is the small high-dimensional dataset we extract the treatment effect from, and it is composed of covariates $X_{target}$, a binary treatment assignment $T \in \{0, 1\}$, and continuous outcome $Y$. The source-domain contains the covariates $X_{source}$, which are from the same feature space as $X_{target}$. Here, we assume the source-domain is unlabeled (without treatment assignment or outcomes). Consider $X = [X_{target}, X_{source}]$, and $n = n_s + n_t$, where $n_s$ and $n_t$ are the sample sizes of the source-domain and target-domain, respectively. We create a new binary variable $D$, which encodes the sample's domain. We define the target-domain set as $\mathcal{D}_S = \{(x_i, d_i = 0) : i = 1, ..., n_s\}$ and $\mathcal{D}_T = \{(x_i, t_i, y_i, d_i = 1) : i = 1, ..., n_t\}$.

The *individual treatment effect* (ITE) for a given sample $i$ is defined as $\tau_i = Y_i(1) - Y_i(0)$, where $Y_i(t)$ is the potential outcomes under $t$. The challenge is each sample $i$ either has a $Y_i(1)$ or $Y_i(0)$ associated, never both. A solution is to work with the *Average Treatment Effect* (ATE), the focus of our work, defined as $\tau = \mathbb{E}[Y(1) - Y(0)]$. We define outcome models as $Y(t) = \mu_t(x) = \mathbb{E}[Y|X = x, T = t]$, and estimate ATE as $\tau = \mu_1(x) - \mu_0(x)$.

### 3.2 Assumptions

In this section, we list the underlying assumptions of Causal-Batle. To illustrate our methodology, we adopt a dragonnet architecture (Shi et al., 2019) as the backbone of our neural network. Other backbones architecture could also be adopted, such as convolutional neural networks for image-based applications.

**Assumption 1**: The target-domain covariates $X_{target}$ and source-domain covariates $X_{source}$ are in the same feature space.

**Assumption 2**: Stable Unit Treatment Value Assumption (SUTVA) (Rubin, 1980) - the response of a particular unit depends only on the treatment(s) assigned, not the treatments of other units.

**Assumption 3:** Ignorability - the potential outcome is independent of the treatments given the covariates

$$\{Y(1), Y(0)\} \perp T | X$$

**Assumption 4:** Positivity/Overlap - the treatment assignment is not deterministic.

$$0 < P(T = t | X = x) < 1$$

**Definition 1**: Back-door Criterion (Pearl, 1995) - Given a pair of variables $(T, Y)$ in a directed acyclic graph $G$, a set of variables $X \in G$ satisfies the backdoor criterion relative to $(T, Y)$ if no node in $X$ is a descendant of $T$, and $X$ blocks every path between $T$ and $Y$ that contains an arrow into $T$.

**Theorem 1**: Identifiability (Pearl, 1995) - If a set of variables $X$ satisfies the back-door criterion relative to $(T, Y)$, then the causal effect of $T$ on $Y$ is identifiable.

Assumption 1 is related to our TL component to ensure both domains share the same features. According to Assumption 2, our target-domain is generated as $(Y_i, T_i, X_i) \overset{\text{i.i.d.}}{\sim} D_{target}$, where $D_{target}$ is the observational distribution of the target-domain. Assumption 2 is also a standard causal inference assumption, which guarantees that one unit does not affect other units. According to Assumption 3, the observed covariates $X_{target}$ contain all the confounders of the treatment $T$ and outcome $Y$. Assumption 3 guarantees that $X_{target}$ will block all back-door paths and satisfy the back-door criterion (Definition 1). Thus, the identifiability of the treatment effect is guaranteed by Theorem 1. Note that we assume a setting with strong ignorability (Assumption 3). If that is not the case, one might need to choose a different backbone architecture and make extra assumptions to guarantee identifiability. However, while there are recent works exploring how to estimate treatment effects with unobserved confounders (Tchetgen et al., 2020; Mastouri et al., 2021; Louizos et al., 2017; Wang & Blei, 2019), they still have several known limitations (Rissanen & Marttinen, 2021; Zheng et al., 2021). Furthermore, working with unobserved confounders is not the focus of this paper.

**Theorem 2**: *Sufficiency of Propensity Score* (Rosenbaum & Rubin, 1983; Shi et al., 2019) - If the average treatment effect is identifiable from observational data by adjusting for $X$, i.e., $ATE = E_X[E_Y[Y|T = 1, X] - E_Y[Y|T = 0, X]]$, then adjusting for the propensity score also suffices: $ATE = E_X[E_Y[Y|f(X), T = 1] - E_Y[Y|f(X), T = 0]]$

According to the Sufficiency of Propensity Score, it suffices to use only the information in $X$ relevant to predicting the propensity score, $P(T = 1|X)$. The propensity score is predicted by $g(.)$, and all the relevant information to predict $P(T = 1|X)$ is the output of $f(.)$. For the proof, please refer to the original publication Rosenbaum & Rubin (1983).

### 3.3 Architecture

Causal-Batle builds upon existing neural network methods that estimate treatment effects. While not tied to any specific architecture, we will explain our methodology using a Dragonnet (Shi et al., 2019) as a backbone. The Dragonnet is a neural network with three heads: one to predict the treatment assignment given the input features, and the other two to predict the outcome if treated and untreated. It assumes no hidden confounders and relies on the Sufficiency of Propensity Score (Theorem 1) to estimate its outcomes,

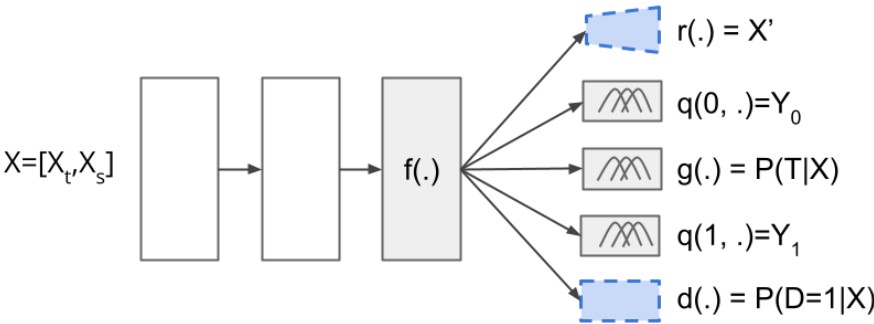

Figure 2: Causal-Batle's architecture when adopting the Bayesian Dragonnet as a backbone architecture. In the blue-traced board, we have our main contributions - the discriminator and the reconstruction heads.

later used to estimate the treatment effects. The Dragonnet architecture has five key components: a shared representation of the input covariates $f(.)$, treatment assignment prediction $g(.)$, a targeted regularization, and the prediction of the outcome if treated and untreated. Jesson et al. (2020) showed that for high-dimensional datasets, Bayesian neural networks tend to yield better results than their non-Bayesian neural network counterpart. They proposed a modification that suits most treatment effect estimators based on neural networks: a Gaussian Process on the last layer.

$$q(Y|T, X, w) = \mathcal{N}(Y|\mu_t(X; w), \sigma_t^2(X; w)) \tag{1}$$

where $Y$, $T$, and $X$ are the observed outcome, the treatment assigned, and the covariates, respectively, and $w$ are the neural network weights.

Up to this point, all components of the architecture address only the estimation of the outcome in a high-dimensional setting. Note that these components assume the sample size $n$ is sufficiently large to train such a neural network. It is also worth noticing that, regardless of the backbone adopted (in our case, Dragonnet), training the shared representation $f(.)$ is the most expensive step: this is the component with the largest number of weights, which receives as input the high-dimensional data and outputs a lower-dimensional representation of that data, used as input by the other components.

Causal-Batle focuses on improving the training of $f(.)$, the hardest part of the architecture. The idea is to use the source-domain to improve the representation of $f(.)$ (hence, transfer knowledge from source-domain to target-domain). With a better trained $f(.)$, the estimation of the outcome models and the propensity score would yield better results.

Causal-Batle adds two new components to the network architecture: a discriminator $d(.)$ and a reconstruction layer $r(.)$, and their corresponding terms in the loss function. The overall architecture is shown in Figure 2, with the two new components highlighted with blue-traced boards. The discriminator component $d(.)$ is a binary classifier defined as $d(f(X)) = P(D = 1|X)$. Therefore, for a given sample $i$, the discriminator will predict whether it belongs to the source or target-domain. The adversarial loss incentivizes $f(.)$ to extract patterns common to both domains and fool $d(.)$. The adversarial loss also helps remove spurious correlations and potential biases present in only one of the domains, improving $f(.)$'s representation. The reconstruction component $r(.)$ is to ensure $f(.)$'s representation is meaningful. For the reconstruction component $r(.)$, in our work, we adopt an autoencoder for simplicity, but we point out that one could replace it with more sophisticated methods, such as variational autoencoders.

Similarly to Jesson et al. (2020), Causal-Batle adopts a Bayesian neural network with a Gaussian Process to estimate the outcomes (See Equation 1), and for the propensity score, with a Bernoulli Distribution:

$$g(T|X, w) = \mathcal{Bernoulli}(T = 1|p(X; w)) \tag{2}$$

Causal-Batle does not assume a distribution for the discriminator $d(D|X)$. Instead, it adopts a neural network classification node, which allows it to use traditional losses for adversarial learning.

In the context of Causal Inference, Causal-Batle components aim to improve the covariate adjustment. Existing causal effect estimators might underperform in applications with small high-dimensional datasets. Often, this underperformance is due to data limitations that prevent the estimator from learning a good feature representation to perform the covariate adjustment. Focusing on these applications (with small high-dimensional datasets), Causal-Battle uses an architecture that allows us to adopt an additional data source, $X_{source}$. By adding another data source, under certain assumptions, the proposed representation would perform a better covariate adjustment, resulting in better treatment effect estimates with a lower variance.

### 3.4 Loss Functions

The exact loss function will depend on the backbone architecture adopted. Here, we will present the loss function for the example given in Figure 2, which is a bayesian dragonnet architecture. Note that we have $n_t$ samples in the target-domain, $n_s$ samples in the source-domain, and $n = n_t + n_s$

There is a loss associated with each grey/blue square. Starting with the outcome model loss: we only observe the outcome $Y$ for either $q(0,.)$ or $q(1,.)$, with $q()$ representing a distribution. Therefore, we define the loss as the log probability:

$$\ell_y(Y, \hat{Y}) = -\frac{1}{n_t} \sum_{i=0}^{n_t} (T_i log(\hat{q}(Y_i|T_i = 1, X_i)) + (1 - T_i)log(\hat{q}(Y_i|T_i = 0, X_i)))] \tag{3}$$

The propensity score loss is also calculated using log probability.

$$\ell_t(T, \hat{T}) = -\frac{1}{n_t} \sum_{i=0}^{n_t} log(\hat{q}(T_i|X_i)) \tag{4}$$

To capture the losses associated with the transfer learning components, we adopt a second binary cross entropy loss for the discriminator. Defining $\hat{D}_i = \hat{d}(X_i)$:

$$\ell_d(D, \hat{D}) = -\frac{1}{n} \sum_{i=0}^{n} (D_i log(\hat{D}_i) + (1 - D_i)log(1 - \hat{D}_i)) = BCE(D, \hat{D}) \tag{5}$$

and to compete with the discriminator loss, we have the adversarial loss:

$$\ell_a(\hat{D}) = \frac{1}{n} \sum_{i=0}^{n} log(1 - \hat{D}_i) \tag{6}$$

Finally, the reconstruction loss ensures the features extracted by $f(.)$ are meaningful. Denoting by $\hat{X}$ the reconstruction of input $X$, we have:

$$\ell_r(X, \hat{X}) = MSE(X, \hat{X}) \tag{7}$$

The final loss is defined as:

$$\ell = \alpha_0 \ell_y(Y, \hat{Y}) + \alpha_1 \ell_t(T, \hat{T}) + \alpha_2 \ell_d(D, \hat{D}) + \alpha_3 \ell_a(\hat{D}) + \alpha_4 \ell_r(X, \hat{X}) \tag{8}$$

where the array $\alpha = [\alpha_0, ..., \alpha_4]$ are the losses' weights.

Fining tuning the losses' weights is extremely important. The most basic approach is to adopt a grid search with a human in the loop to find each component's minimum loss and convergence. The convergence is

measured by tracking the losses per epoch and comparing the losses on the training and validation set. While simple, this approach can be time consuming. For advanced users, a task balancing loss approach, such as Dynamic Weight Average (DWA) Liu et al. (2019) could also be adopted after suitable adaptations.

### 3.5 Treatment Effect Estimation

Equation 9 shows how the treatment effect is estimated. To calculate the Average Treatment Effect (ATE), we use the mean value $\mu_t(x, w)$ of the Normal Distribution (see Equation 1). The ATE can then be estimated as:

$$\hat{\tau} = \frac{1}{n} \sum_{i=0}^{n} [\mu_1(x_i, w) - \mu_0(x_i, w)] \tag{9}$$

### 3.6 Source-domain and Target-domain relationship

This section discusses the relationship between the source and target-domain. Causal-Batle aims to use the unlabeled source-domain to learn a better representation of the labeled target-domain. There are three losses associated with the transfer learning: the discriminator loss $\ell_d$, the adversarial loss $\ell_a$, and the reconstruction loss $\ell_r$. Here, we discuss the impact of $P(X_{target})$ (target-domain distribution) and $P(X_{source})$ (source-domain distribution) in our proposed methodology.

- $P(X_{target}) = P(X_{source})$: $\ell_d$ will be large, as the discriminator will have a hard time trying to distinguish elements from two identical distributions while working against the adversarial loss. The $\ell_r$ loss, on the other hand, will thrive. A large collection of samples from the same distribution will help $f()$ to find a good representation of the input data, lowering the $\ell_r$.

- $P(X_{target}) \neq P(X_{source})$: $\ell_d$ will be low, as $d(.)$ would be able to identify these differences and use them. Still, $\ell_a$ would attempt to fool $d(.)$, which would push $f(.)$ to learn a domain agnostic representation.

This means that, in practice, regardless of the $P(X_{target})$ and $P(X_{source})$ distributions, the losses would push $f(.)$ to have a meaningful representation of $X$. The key is to find a balance between the adversarial loss, the discriminator loss, and the reconstruction loss, which, as shown in Equation 8 is controlled by the weights (hyper-parameters) $\alpha$. Hence, the losses' weights, when well calibrated, contribute to avoid the degradation of the performance when the distributions are different.

## 4 Experiments

Our experiments compare Causal-Batle against some of the most adopted causal inference baselines for binary treatments: Dragonnet (Shi et al., 2019), Bayesian Dragonnet (Jesson et al., 2020), AIPW (Glynn & Quinn, 2010), and CEVAE (Louizos et al., 2017). Note that none of these baselines adopt transfer-learning techniques. The AIPW was adopted in its simplest form with two linear regressions as base-models and logistic regression as propensity score. As future work, it would be very interesting to plug-in the results of our neural network as a base-model of a AIPW.

The main questions we want to investigate are:

- Does the transfer learning approach improve the treatment effect estimates compared to existing treatment effect methods?

- What is the impact of the ratio between the sizes of $X_s$ (unlabeled source-domain) and $X_t$ (labeled target-domain)?

To make a fair comparison between Causal-Batle and the baselines, we adopt the following scheme:

**Setting 1** - Datasets with only $X_{target}$: In the evaluation, we also adopted traditional benchmark datasets, which do not have an unlabeled source domain. To adapt to our setting, we splited the $n$ samples of $X_{target}$ into two sets: $n \times p_t$ samples to $X'_{target}$ and $n \times (1 - p_t)$ samples to $X'_{source}$. The quantity $p_t$ represents the proportion of the dataset used as target-domain, with $0 < p_t < 1$. We removed the labels (treatment assignment and outcome) from the samples in $X'_{source}$. The baselines receive the labeled target-dataset $X'_{target}$, and Causal-Batles receives the target-dataset $X'_{target}$ and the unlabeled source-dataset $X'_{source}$.

**Setting 2** - Datasets with $X_{source}$ and $X_{target}$: Baselines are trained using only $X_{target}$. Causal-Batle would have access to both domains.

We define the ratio as $r = n_t/n_s$, representing the target-domain's size relative to the source-domain. The Causal-Batle ideal use-cause is for applications with low $r$. The code for an implemented version of Causal-Batle is available at `github/HiddenForDoubleBlindSubmission`. We simulated each dataset $b_d$ times and fit each method $b_m$ times. Therefore, each given combination of (dataset × method) was repeated $b_d \times b_m = B$ times for consistency and reproducibility. For the Causal-Batle, Dragonnet, and Bayesian Dragonnet, we adopted the same backbone architecture to have a fair comparison. As metric, we adopt the Mean Absolute Error (MAE) between the estimated treatment effect $\hat{\tau}$ and the true treatment effect $\tau$, defined as $MAE = |\hat{\tau} - \tau|$. In our plots and tables, we report the average value over all the $B$ repetitions: $\overline{MAE} = \frac{\sum^B MAE}{B}$. Low values of MAE are desirable. The experiments were run using GPUs on Colab Pro+.

## 4.1 Datasets

We adopted the three datasets to validate our proposed method. Note that we adapted these datasets to suit the application we want to mimic, which contains small high-dimensional datasets.

### 4.1.1 GWAS

The Genome-Wide Association Study (GWAS)(Wang & Blei, 2019; Aoki & Ester, 2021) dataset is a semi-simulated dataset that explores a high-dimensional sparse setting. Proposed initially to handle multiple treatments, we adapt it to have one binary treatment $T$ and a continuous outcome $Y$. We split the data into two datasets, as explained in Setting 1. Note that $X_{source}$ and $X_{target}$ have the same distribution. We simulated a small high-dimensional dataset with 2k samples and 10k covariates.

The core idea is to simulate covariates that mimic the behavior of single-nucleotide polymorphisms (SNPs), and the outcome represents a clinical trait. This dataset allows the simulation of single and multiple binary treatments, and the process described below is the same adopted by Wang & Blei (2019). The 1000 Genome Project (TGP)Consortium et al. (2015) is used to make the simulations more realistic. The first step is to generate this dataset is to use Linkage Disequilibrium to remove TGP's highly correlated SNPs, and adopt a PCA to extract $c = 2$ components. The resulting dataset is the genetic representation matrix $\Gamma_{V,c}$, where $V$ is the number of remaining covariates. An interception column $\Gamma_{V,c+1} = 1$ is appended to $\Gamma$. The patients' representation matrix is generated as $\Pi_{N,c} \sim 0.9 \times Uniform(0, 0.5)$, where $N$ is the number of desired samples. The last column of the patient's representation matrix is constant $\Pi_{N,c+1} = 0.05$ (intercept column). The product of these two matrices, $\Pi_{N,c} \times \Gamma^T_{V,c}$, represents the allele frequency, later used to simulate the covariates $X$. The covariates are simulated as $X_{N,V} \sim Binomial(1, \Pi_{N,c} \times \Gamma^T_{V,c})$. The effect of the treatment is defined as: $\tau \sim Normal(0, 0.5)$. To calculate the outcome, three groups were extracted using k-means($X$) to add confounding. Each group $l \in \{1, 2, 3\}$ has an intercept value $\lambda_l$ and noise distribution $\epsilon \sim Normal(0, \sigma_l)$, $\sigma_l \sim InvGamma(3, 1)$. Following a high signal-to-noise ration, the SNP's and the per-group intercept are responsible for 40% ($v_{gene} = v_{group} = 0.4$) of the variance each, and the error is responsible for 20% ($v_{noise} = 0.2$) of the variance. To re-scale the noise and intercept:

$$\lambda_{\leftarrow} \left[ \frac{sd\{\sum_V \tau_v X_{n,v}\}^N_{n=0}}{\sqrt{v_{gene}}} \right] \left[ \frac{\sqrt{v_{group}}}{sd\{\lambda\}^N_{n=0}} \right] \lambda \tag{10}$$

$$\epsilon_j \leftarrow \left[ \frac{sd\{\sum_V \tau_v X_{n,v}\}_{n=0}^N}{\sqrt{v_{gene}}} \right] \left[ \frac{\sqrt{v_{noise}}}{sd\{\epsilon_j\}_{n=0}^N} \right] \epsilon_j \tag{11}$$

Finally, the outcomes are generated as:

$$Y = \tau X_{n,t} + \lambda_{c_j} + \epsilon_j \tag{12}$$

### 4.1.2 IHDP

The Infant Health and Development Program (IHDP)Hill (2011) is a traditional benchmark to evaluate treatment effect estimators that target single binary treatments. It simulates data to mimic a study on infant development. In that study, the treatment was assigned ($T = 1$) if the child had special care/home visits from a trained provider. The outcome $Y$ is cognitive test scores, and the goal is to measure the causal effect of the home visits. The benchmark contains ten replications of such a study, with 24 covariates and a continuous outcome. We adopt this dataset to compare our proposed method with some of the single-treatment baselines that have been previously evaluated on the IHDP benchmark datasets [1]. This dataset has been adopted by several of our baselines. We adopted Setting 1. Note that this dataset is not high-dimensional. Hence, the IHDP is not the ideal use case of our proposed method; still, it is a classic dataset in Causal Inference, so we adopted it to compare with existing methods.

### 4.1.3 CMNIST

A semi-synthetic dataset that adapts the MNIST (LeCun, 1998) dataset to causal inference. It contains a binary treatment and continuous outcome. We follow the setup from Jesson et al. (2021) with small adaptations to contain a source and a target-domain. Jesson et al. (2021) generates the treatment assignments $T$ and the outcomes $Y$ partially based on the images' representation. In our adaption, we also follow this scheme, but we only used two randomly selected digits $c_i, c_j \in \{0, .., 9\}, c_i \neq c_j$. Therefore, in our target-domain, we have the images of the digits $c_i$ and $c_j$. We map the high-dimensional images to a one-dimensional array $\phi$, which is used to simulate the treatments $T$ and the outcome $Y$ as follows. We calculate the average intensity of the images with the digits $c_i$ and $c_j$, $\mu_c i$ and $\mu_c j$, and standard deviations $\sigma_c i$ and $\sigma_c j$. The average intensity of a sample $i$ is defined as $\overline{X}_i$. The high-dimensional images $X$ are mapped into a one-dimensional manifold $\phi \in [-2, 2]$ with a linear transformation defined by $[Min_c, Max_c]$.

$$\phi = \left( Clip \left( \frac{\overline{X} - \mu_c}{\sigma_c}; -1.4, 1.4 \right) - Min_c \right) \frac{Max_c - Min_c}{1.4 - (-1.4)} \tag{13}$$

The treatment assignment is generated as:

$$T \sim Bernoulli(Sigmoid(2\phi + 0.5)) \tag{14}$$

The outcome is generated as:

$$Y = (2T - 1)\phi + (2T - 1) - 2Sin(2(2T - 1)\phi) + 2(1 + 0.5\phi) + \epsilon$$

where $\epsilon \sim N(0, 1)$.

For the source-domain, we use the images of the other digits. As the source-domain is unlabeled, there is no treatment $T$ or outcome $Y$ produced, and we use the images as they are observed. We follow Setting 2.

### 4.2 Experimental Results

**Setting 1 (Figure 3A-D)**: We explored the following ratios $r \in \{0.25, 0.67, 1.5, 4\}$, with low $r$ being the ideal use-case of Causal-Batle. For the IHDP dataset, we adopted $b_d = 10$ datasets replications with $b_m = 30$ model repetitions, for a total of $B = 10 \times 30 = 300$ estimated values for each $r$. For the GWAS dataset,

---

[1]IHDP dataset is available at `github.com/AMLab-Amsterdam/`

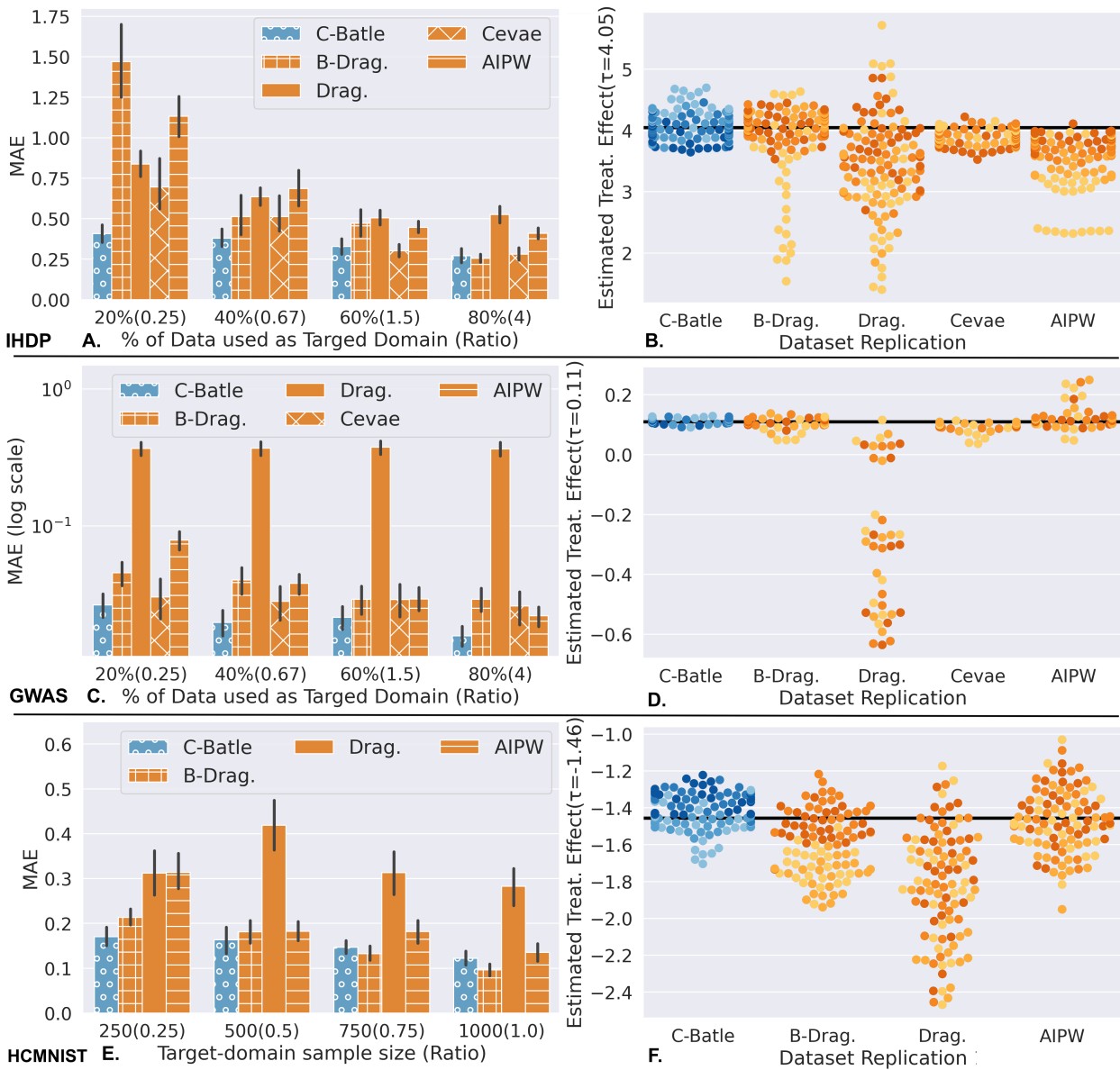

Figure 3: Experimental Results. Plots A, C and E show the average MAE of the IHDP, GWAS and HCMNIST datasets, and the black vertical line is an error bar (95% Confidence Interval). Plots B, D and F show estimated values of individual runs on a dataset replication of the IHDP, GWAS and HCMNIST datasets. The black horizontal line indicates the true treatment effect, and the color gradient encodes the ratio $r = n_t/n_s$ (light colors $\rightarrow$ low $r$ , darker colors $\rightarrow$ high $r$).Legend: C-Batle: Causal-Bate, B-Drag.: Bayesian Dragonnet, Drag.: Dragonnet.

we adopted $b_d = 10$ dataset replications with $b_m = 10$ model repetitions for each $r$ ($B = 100$). The IHDP dataset is not a small high-dimensional dataset; however, it is a classic benchmark dataset. With $r \leq 1$, Causal-Batle outperforms all the other baselines as shown in Figure 3.A. With $r > 1$, the performance of all methods improves, including Causal-Batle, which stays in the top 2. Figure 3.B shows the $b_m = 30$ model repetitions on a dataset replication, where the colors indicate the $r$ value (lighter colors $\rightarrow$ low $r$). Besides having the lowest MAE's, Causal-Batle estimations are well distributed around the true value, while some baselines often underestimate the true values, especially for low $r$ values (light-colored dots).

The GWAS dataset fits more in our ideal use-case, containing 2k samples and 10k covariates. As a reminder, $p_t = 0.2(20\%)$ means that all methods will receive $2k \times 0.2 = 400$ samples as $X_{target}$, which contains the treatment assignment and the outcome of interest; Causal-Batle, on top of the $X_{target}$, also receives $X_{source}$, which contains $2k \times (1-0.2) = 1.6k$ unlabeled samples (only covariates available, without treatment assignment nor outcome). Figure 3.C shows that Causal-Batle has the lowest MAE among all methods considered, with the distance between Causal-Batle and the other methods decreasing as $r$ increases. Figure 3.D shows that Causal-Batle estimated values are well distributed around the true value. Note that the Bayesian Dragonnet and CEVAE are also well distributed around the true value; however, they underestimate $\hat{\tau}$ with low $r$ values.

**Setting 2 (Figure 3E-F)**: The HCMNIST had $b_d = 4$ dataset replications and $b_m = 25$ model repetitions, for a total of $4 \times 25 = 100$ estimated values for each $r \in \{0.25, 0.5, 0.75, 1\}$. As Figure 3.E shows, Causal-Batle has the smallest MAE for $r \leq 0.5$, and the second-best MAE for $r > 0.5$. According to the experimental results in Figure 3.F, both Causal-Batle and the Bayesian Dragonnet tend to underestimate the treatment effect with low $r$ values (light colors). However, Causal-Batle has lower variance, and its estimates are more accurate than Bayesian Dragonnet's estimates.

**Discussion:** Causal-Batle has more accurate estimates (low MAE) than the baselines for low $r$ values. Note that the only difference between Causal-Batle and the Bayesian Dragonnet is the transfer learning components (discriminator loss, adversarial loss, reconstruction loss, and the use of the $X_s$ dataset). An explanation for these results is that transfer learning improves the estimates. Considering the IHDP dataset, when $r > 1$, the simpler architecture of the Bayesian Dragonnet performs better, indicating a possible threshold for the IHDP dataset when a simpler architecture is preferable over a more complex one with transfer learning. For the GWAS and HCMNIST datasets, that threshold is above $r > 4$ and $r > 0.5$. Note that we did not experiment with CEVAE with the HCMNIST as that would require many changes to the CEVAE to work with image data, which was not our focus. We also investigated Dragonnet's poor performance. Our main hypothesis is that the Bayesian component included in the Bayesian Dragonnet and Causal-Batle reduces the variability and improves $\tau$ estimates. The Bayesian component predicts the outcome and the treatment assignment as a Gaussian Mixture and Bernoulli distribution, respectively, and predictions are calculated using MC-dropout (30 forward passes).

## 5 Conclusion

This paper addresses an important and underexplored problem: causal inference in small high-dimensional datasets. In Section 2 we describe existing approaches to estimate treatment effects with a binary treatment and continuous outcome, along with their main limitations. We later describe in Section 3 our assumptions and our proposed method, Causal-Batle, and the main characteristics of the target and source-domain datasets. Finally, in Section 4 we validate our method, comparing it with existing methods in three different datasets. The Causal-Batle ideal use case is tested in our experiments when we have low $r = n_t/n_s$. Under this setting, Causal-Batle has the best performance among all the baselines considered. In other settings, with larger target-domain datasets, the performance is comparable to the baselines. Therefore, according to our experimental results, the transfer learning components of Causal-Batle improved the treatment effect estimations for low values of $r$ compared to other methods.

Causal-Batle, like other machine learning methods, could have negative societal impacts if misused, fed biased data, or applied to unethical applications. In general, we believe most Causal-Batle negative characteristics are inherent to the field (of machine learning) rather than our proposed method. Ethics guidelines can help reduce unethical applications while the others can be mitigated by increasing the transparency in methodologies, datasets, and code availability. Causal-Batle's dependence on the availability of an unlabeled source-domain dataset in the same feature space as the target-domain dataset can be considered a limitation of our approach, along with the strong ignorability assumption that is often difficult to guarantee in real-world applications. To address the first limitation, we recommend checking the baselines adopted by our work, in particular the Bayesian Dragonnet(Jesson et al., 2020) and CEVAE(Louizos et al., 2017), as these two methods do not need external source-domain datasets. As for the strong ignorability assumption, one might

adopt a sensitivity analysis (Veitch & Zaveri, 2020) to investigate the impact of unobserved confounders on the estimation of treatment effects.

Overall, Causal-Batle demonstrated to be a good estimator for its intended use case. Such a setting is very common in Computational Biology applications, where we often have high-dimensional datasets with genetic information but a limited number of samples available with treatment assignment and observed outcome. Yet, several unlabeled datasets in the same feature space could be used under certain assumptions to improve causal inference in small high-dimensional datasets. Another use-case example is when images are used as covariates to perform causal inference - a topic that has recently received more attention. In this case, if only a few labeled images are available to estimate the treatment effects, one could adopt several unlabeled (similar) images to improve the feature extraction component of Causal-Batle, and, consequently, improve the treatment effect estimation.

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
