# OpenReview forum: "Causal Inference from Small High-dimensional Datasets"
_TMLR — Rejected by TMLR_

### Review · Reviewer_yJcN · 2022-10-07

**Summary Of Contributions:**

This paper constructed a treatment effect estimator based on the Dragonnet and Bayesian Neural Network techniques when the source and target domains are different. Multiple experimental results advocate the proposed estimator.

However, I can't entirely agree with the claimed contribution. I don’t think the proposed estimator successfully tackles the transfer learning problem because the paper only considered the setting where the source and the target domains are the same. Moreover, the current shape of the paper contains many grammatical and mathematical errors and confusion. Finally, I have some issues and questions about the experimental results.

**Broader Impact Concerns:**

I don't think that the paper has any societal issues.

**Requested Changes:**

** Title and the model's name **
1. Please consider changing the title — It seems that the paper tackles the domain adaption problem, but it's not mentioned in the title.

2. Please consider changing the name of the model — It should be Causal BATLE instead of Causal Batle, because "Batle" isn't an English word (up to my knowledge).

** Points to be improved on writing **
1. For example, In Sec. 3.1, "The target-domain is the small high-dimensional dataset" — Is the domain the dataset? I don't think that "domain" in the source-domain or the target-domain means the dataset. Instead, it means the data generating process / distribution / population of the dataset.
2. In Section 2, what does "Target (Shalt et al., 2017)" means?
3. Please improve the resolution of the Figures.
4. In the 2nd paragraph of Section 1, "Nevertheless, these methods do not adopt typical transfer learning approaches from the ML community". This sentence could be offensive to many stat-ML researchers because it reads that any methods not adopting the ML technique are bad, which is not true at all.
5. In the last sentence of the 2nd paragraph, how does "mentioning causality" improve the transfer learning method?
6. In Section 3.1., when explaining the ITE with Y_i and the ATE with Y, there must be an assumption that samples are iid. Otherwise, Y_i and Y must be separately defined.
7. Please explain the Dragonnet and Bayesian Dragonnet (Figure 2) as preliminaries.

** Points to be improved on math **
1. Assumption 1 must be mathematically formalized because the "same feature space" is equivocal. For example, it could either mean that two $X_{target}$ and $X_{source}$ follow the same distribution or two covariates are in the L_2(P) space, which leads to the solution being completely different.
2. Assumption 3 and the Back-door criterion are actually the same assumptions. Therefore, you don't need to mention the back-door criterion when Assumption 3 is declared.
3. More specifically, when you assume that the causal graph is given and the distribution is compatible with the graph, then the back-door can be used as an assumption. Without the graph, however, it’s natural to go by Assumption 3. Given that the paper doesn’t contain any graphs, I think you don’t need to mention the back-door criterion and the identifiability.
4. It’s mentioned that Eq. (1) is the treatment effect estimator. However, it seems that Eq. (1) is a conditional distribution of Y given T,X and w. It needs more explanations why Eq. (1) means the treatment effect.
5. How can Equation 9 means the ATE? The ATE is not dependent on the value of X=x. However, Eq. (9) depends on X=x. I think you might intend the ITE.
6. How come the Equation 1 is a Gaussian Mixture? Isn’t it just a Gaussian distribution? It’s even mentioned in the paper (Section 3.5) that Equation 1 is the Gaussian distribution.

** On experimental results **
1. It seems that the proposed estimator is highly dependent on the performance of q(Y | T,X,w). However, the AIPW estimator has the doubly robust property — It converges much faster than either q(Y | T,X,w) and the g(T | X,w), and it’s consistent even when either q or g is wrongly estimated. Therefore, it doesn’t look right to me that the proposed estimator is significantly outperforming the AIPW estimator.
2. Is the Bayesian Neural Network sample-efficient compared to the simple and classic estimator? If it’s not the case, I think the title “small dataset” is not proper.



**Strengths And Weaknesses:**

# Strengths
This paper conducted multiple simulations on various datasets.

# Weakness
1. As mentioned in the "Summary Of Contribution", I'm afraid I have to disagree with the claimed contribution. I think the imposed assumption that the domain and the target are in the same feature space is too strong to make the problem somewhat trivial. Under such an assumption, the standard one-domain-based estimator can be used because the target domain can be understood as a test dataset, where the trained model can be naively applied without any special treatises. Since no special techniques in transfer learning are required in this setting, I think the problem is not much different from the one-domain problem. Therefore, claiming 'domain adaption' in such a setting is somewhat overclaiming.

2. Please consider changing the title — It seems that the paper tackles the domain adaption problem. Still, it’s not mentioned in the title.

3. Please consider changing the name of the model. It should be Causal BATLE, instead Causal Batle, because “Batle” isn’t an English word (up to my knowledge).

4. I think the writing can be improved significantly. The current form of the paper contains many grammatical errors and logical gaps between sentences. Here are a few examples. Note that there are much more errors and gaps other than the ones mentioned in the "Requested Changes" section.

5. I think the mathematical formality could be significantly improved. In the current shape, there are duplicated assumptions (back-door and ignorability), equivocal notions (the same feature space), errors (”Gaussian mixture” in Eq. (1)), omitted explanation (”Bayesian Dragonnet vs. Draggonnet”), and confusion (ITE vs. ATE).

6. An explanation of the architecture is insufficient. To my understanding, the proposed method is the Dragonnet improved by the Bayesian Neural Network technique. If it’s right, please write it more explicitly, and explain the architecture more transparently.

7.  I have doubts and questions about the experimental results. Specifically, I am not sure how the proposed estimator can outperform the AIPW estimator, given that the AIPW is provably a better estimator when the ML methods are combined.

---

### Review · Reviewer_m8a2 · 2022-10-24

**Summary Of Contributions:**

The main contribution of the paper is a method akin to transfer learning that helps learn a representation to improve causal estimation on a small dataset with outcomes, confounders, and covariates all observed. The key idea is to simultaneously use a small labelled dataset with $Y,T,X $ observed and a large data set with only $X$ observed. Then, the claim is that the task of learning the conditional $Y \mid T, X$ can be made efficient by
1. learning a representation $f(X)$ from large dataset of just $X$
2. Such that $(Y_1, Y_0) \perp T \mid  f(X)$ and
3. the error in learning $Y\mid T, f(X)$ from the small labelled dataset is smaller than the error in learning $Y\mid T, X$.


This idea seems novel and allows for using large unlabelled data which makes it immediately useful on tasks like genetics where typically $d>>n$ for datasets with $Y,T,X$ all observed.

**Broader Impact Concerns:**

No additional concerns beyond the potential negative uses of any causal inference method.

**Requested Changes:**


As I see it, there are two cases where the method may not help or make estimation worse. Both stem from requiring that $f(X)$ helps model the marginal distribution of $X$, which is an orthogonal task to modeling $Y \mid T, X$ well, in general.

1. When the input covariates cannot be compressed into a sufficiently low dimensional representation: for example if the covariates are sampled from an isotropic gaussian, $X\sim \mathcal{N}(0, I_{d\times d})$ but $E[Y \mid T, X]$ only depends on the first dimension $X_1$. Then Causal-Batle will ask the function $f(X)$ to help model all the dimensions of $X$, most of which do not help in the causal estimation problem.


2. When the the function $g(X) = E[Y \mid T=1, X]$ is a much simpler than $\hat{g}(f(X)) = E[Y \mid T, f(X)]$ . This can make estimation worse when modeling $X$ using $f(X)$ then leads to learning a more complex function from the small labelled dataset.

It would be useful to have a discussion of these cases in the paper. In either cases, it is possible that hyperparameter tuning can avoid making the estimation problem worse. If the authors can point out how one can tune the parameters to avoid these issues, that would make the paper better.


One possible investigation that could present the limitations even better is to introduce one of the above limitations into an experiment. My suggestion would be the following empirical evaluation: add additional dimensions to the covariates that are just noise, and see how the method behaves as the added dimension is made to be large.  This also fixes an issue the authors mention: "Note that this dataset is not high-dimensional. Hence, the IHDP is not the ideal use case of our proposed method".



While a theoretical upper bound on error that can be shown to be smaller than when using the labelled dataset alone would be useful, it might require a much larger change to the paper than I can reasonably ask the authors now. So, I would be happy with an intuitive discussion without the theoretical support.


**Strengths And Weaknesses:**

The presentation of the method is good and the experiments look promising, showing improved results on a synthetic GWAS, IHDP, and a MNIST-based causal estimation task.


My main concerns are two fold:
1. The limitations of where the method should be used are discussed well enough.
2. There 4 hyperparameters in the method; the authors have not discussed tuning these.

2 becomes important due to potential mis-matches between the best $f(X)$ is for modeling $X$ and the best $f(X)$ for modeling $E[Y \mid T, X]$. See requested changes for questions and further details.

---

> ### Author Response · Authors · 2022-10-31
> **Comments on requested changes**
>
> Thanks for your review!
> We thank you for pointing out that the discussion on the hyperparameters needed to be more explicit. As you correctly noticed, tuning these parameters can be used to avoid the issues mentioned by you. We discussed the tuning at a high level in Section 3.6, and we are happy to extend it and extend the discussion on how to tune them. On the topic of upper bounds, adding theoretical upper bounds on error would be very useful. However, we see that as future work. Still, we will add a short discussion on the conclusion.
>
> The experiment mentioned is partially covered with the HCMNIST dataset. Note that, for that dataset, the target domain contains digits not present in the source domain. We agree that this example is less extreme than an experiment with covariates that are just noise. Yet, it helps to illustrate a scenery where the distribution of the target and source domains are not the same. Still, the experiment suggested is reasonable and would provide good insights into the paper’s limitations. If, after adding the other suggestions made by you and other reviewers, we still have space on the paper, we are happy to add this extra experiment.

---

### Review · Reviewer_EnFo · 2022-11-02

**Summary Of Contributions:**

The paper considers a setting when the data set is small-size and high dimensional for applying backdoor criterion. An additional unlabeled source data set is supposed to be available. The source data help learn a better covariates representation with the proposed adversarial discriminative regularization and a reconstruction regularization.

**Requested Changes:**

Please refer to the major concerns. The expected changes include

* Explain why the proposed representation desiderata are beneficial for the goal of causal inference
* Make the heuristic claims concrete
* Clearly state the assumptions on X_{source}
* Offer guidance to practitioners
* Discuss the bias-variance trade-off and when (if not always) the data fusion is preferred

**Strengths And Weaknesses:**

Strength:
The paper is written in a clear way. The small sample problem is well-motivated. It is important to develop methods that can borrow strength from multi-source data for causal inference. Figure 2 and Eqs 3-8 give a clear explanation of the proposed Causal-Batle method. Simulations show the advantage of the proposed method especially when the target data set is small.

Major concerns:

1. It is unclear why the proposed representation desiderata are beneficial for the goal of causal inference. The adversarial discriminative objective has been shown to be useful for domain adaptation, and the reconstruction objective is useful for generative models. But both the domain adaptation methods and generative models are based on statistical associations. Moreover, there are some discussions that the adversarial domain adaptation objective is not aligned with the goal of causal inference (e.g. https://arxiv.org/pdf/1907.02893.pdf).

2. Related to the point above, some claims in the paper are at most heuristic (e.g. "With a better trained f(.), the estimation of the outcome models and the propensity score would yield better results", "the reconstruction loss ensures the features extracted by f(.) are meaningful". "The adversarial loss also helps remove spurious correlations and potential biases present in only one of the domains, improving f(.)’s representation. The reconstruction component r(.) is to ensure f(.)’s representation is meaningful".) These are the key explanations of why using the new objective, but it is unclear what is the exact meaning of "better", "meaningful", and "spurious correlations", and why these claims are true mathematically.

3. Another concern is the unclear assumption over the source data X_{source}. The discussion in Sec. 3.6. needs to be more concrete. Clearly, not all X_{source} will lead to improved errors. Maybe the author can consider: (i) if p(X_{source}) = p(X_{target}), is there a guaranteed improvement by applying the proposed method? (ii) if p(X_{source}) \neq p(X_{target}), what invariance property should be maintained over the two domains (e.g. the same causal graph or a shared part of SEM)? Should p(X_{source}) satisfy any assumptions such as the ignorability?

4. The practical usefulness of Causal-Batle may be limited. First, it is unclear what X_{source} a practitioner should use. Second, there are 4 hyperparameters that need to be tuned in the final objective, which could be hard to find the optimal one (the simulations didn't talk much about how to choose hyperparameters).

5. The paper makes a bias-variance trade-off. The variance problem in the original small data sets scenario is traded for a bias problem due to concerns 1-4 above. It is not convincing that such a trade-off is worthwhile.

Minor:
1. The adversarial object seems directly related to the adversarial discriminative domain adaptation (https://arxiv.org/pdf/1702.05464.pdf) but somehow it is not cited.

2. Pages 4, "We define the target-domain set" --> "We define the source-domain set"

---

> ### Author Response · Authors · 2022-11-09
> **Clarifications and comments**
>
> Thanks for your review! Here are some clarifications and items we commit to improve on the current paper:
>
> - _Explain why the proposed representation is beneficial for the goal of causal inference_: Our proposed work targets the covariate adjustment. In causality, covariate adjustment is used to close back doors and reduce the variance of the treatment effect estimation. Traditional methods with small high-dimensional datasets might underperform in learning a good representation due to data limitations. Note: existing causal inference methods will still perform the covariate adjustment, but the variance might not be as low as it could be. Similarly, the causal effect estimates might not be as accurate as they could be due to the dataset limitations. Focusing on these applications (with small high-dimensional datasets), Causal-Battle uses an architecture that allows us to adopt an additional source of data, $X_{source}$. By adding another data source, the feature extraction with the proposed representation would perform a better covariate adjustment. A more accurate representation is beneficial for the goal of causal inference of obtaining unbiased estimates of the treatment effect with a lower variance. We will modify the introduction to have this contribution more clearly stated.
>
> - We are expanding the discussion in Section 3.6 to clarify the assumption over the source data $X_{source}$. Our method, under the cited assumptions, provides better results than baselines when $P(X_{source}) = P(X_{target})$. If the distributions are different, the loss weights (hyperparameters), when well-calibrated, help to avoid the degradation of the performance by decreasing the weights of the reconstruction loss and increasing some of the other weights. We are already working on clarifying this part in the text. As to clarify your question, X_{source} and X_{target} share the same underlying causal graph; however, $X_{source}$ does not contain a treatment assignment or an outcome.
>
> - The hyperparameters are also connected with the biases-variance trade-off, which is this present in our experiments (See Section 4.2). We didn’t explicitly describe the experiments as a biases-variance trade-off. Still, the intuition behind some of the analyses is very similar. We used the ratio $r=size(X_{source})/size(X_{target})$ to help interpret some of the results in terms of the dataset size and complexity of the NN adopted. As discussed in Section 4.2, there is a threshold where adopting the $X_{source}$ and increasing the architecture complexity with the reconstruction term and adversarial training is not worthed. The experiments also show several instances where adding the $X_{source}$ improves the results. This is often the case with small high-dimensional datasets (the main application target of our proposed method). When working with these datasets, adopting X_{source} and Causal-Batle provides much better results than the alternatives that contain only $X_{target}$.
>
> - Considering the practical usefulness: Causal-Batle has several potential uses in biomedical applications. In these applications, labeled data with treatments and outcomes might be expensive or hard to obtain, yet, unlabeled $X_{source}$ data might be available. Here is an additional example of an application where our proposed method could be adopted: consider a study that wants to investigate a certain intervention $T$ on eye pressure $Y$. The covariates available are ​​fundus images (an image of the rear of the eye). The dataset of such a study could be augmented by adding other fundus images as $X_{source} $(images without treatment assignment or eye pressure associated) to help better train the NN that will extract the feature representation.
>
> - Considering the hyperparameters: we will extend the discussion on how to better tune them. The most basic approach is to adopt a grid search with a human in the loop to find each component's minimum loss and convergence. The convergence is measured by tracking the losses per epoch and comparing the losses on the training and validation set. For advanced users, task balancing loss, such as Dynamic Weight Average (DWA, Liu, Shikun, Edward Johns, and Andrew J. Davison. "End-to-end multi-task learning with attention." Proceedings of the IEEE/CVF conference on computer vision and pattern recognition. 2019.), could also be adopted. We will add this in more detail in the paper.

---

### Decision · Action_Editors · 2022-12-27

**Recommendation:** Reject

**Comment:**

The reviewers raise important concerns on the definition of the scope of the method and the lack of transparent treatment of hyperparameters that need to be addressed for the paper to be publishable. The discussion in Sec. 3.6 is a start on the first, but so far seems to focus more on wishful thinking than significant evidential backing.

In addition to the comments raised by the reviewers, I would encourage the authors to seriously consider the relation of their work to semi-supervised learning, as their setting of using additional unlabelled data to complement smaller labelled data appears to match the standard semi-supervised setting.

In terms of whether such additional unlabelled data is useful, I would strongly encourage the authors to evaluate their work in relation with the classic "On Causal and Anticausal Learning" (B. Schölkopf et al., ICML 2012).

**Audience:**

Assuming the findings are valid, they would likely be of interest to some readers. Given the open questions on the validity, this is not as clear.

**Claims And Evidence:**

The paper claims to present a method that applies transfer learning for semi-supervised causal learning: improving performance by using a larger unlabelled data set to help learning. The authors make very broad claims that the method improves or at least does not hurt performance under very general conditions. The evidence in support of these claims consists of empirical evaluation with three data sets. Given the limited scope of these data sets relative to the very broad claims and lack of transparent algorithm description (especially the description of hyperparameter selection seems lacking), I do not consider there is enough evidence to support the claims.